# Multi-context genetic modeling of transcriptional regulation resolves novel disease loci

Mike Thompson [1] ✉, Mary Grace Gordon[2,3,4], Andrew Lu[5], Anchit Tandon[6], Eran Halperin[1,7,8,9], Alexander Gusev[10,11], Chun Jimmie Ye [2,3,12,13,14], Brunilda Balliu[9] & Noah Zaitlen [1,15] ✉

A majority of the variants identified in genome-wide association studies fall in non-coding regions of the genome, indicating their mechanism of impact is mediated via gene expression. Leveraging this hypothesis, transcriptome-wide association studies (TWAS) have assisted in both the interpretation and discovery of additional genes associated with complex traits. However, existing methods for conducting TWAS do not take full advantage of the intra-individual correlation inherently present in multi-context expression studies and do not properly adjust for multiple testing across contexts. We introduce CONTENT—a computationally efficient method with proper cross-context false discovery correction that leverages correlation structure across contexts to improve power and generate context-specific and context-shared components of expression. We apply CONTENT to bulk multi-tissue and single-cell RNA-seq data sets and show that CONTENT leads to a 42% (bulk) and 110% (single cell) increase in the number of genetically predicted genes relative to previous approaches. We find the context-specific component of expression comprises 30% of heritability in tissue-level bulk data and 75% in single-cell data, consistent with cell-type heterogeneity in bulk tissue. In the context of TWAS, CONTENT increases the number of locus-phenotype associations discovered by over 51% relative to previous methods across 22 complex traits.

A large portion of the signal discovered in genome-wide associations studies (GWAS) has been localized to non-coding regions[1]. In light of this, researchers have developed post-GWAS approaches to elucidate the functional consequences of variants and their impact on the etiology of traits[2]. One notable approach has been to generate genetic predictors of gene expression and leverage these predictors with GWAS data to associate genes with traits of interest[3,4]. These transcriptome-wide association studies (TWAS) have not only shown great promise in terms of discovery and interpretation of association signals but have also helped prioritize potentially causal genes for complex diseases[5]. Nonetheless, methods like TWAS are limited by the

accuracy and power of the genetic predictors generated in training datasets[6–11].

The original TWAS methodology builds genetic predictors of expression on a context-by-context basis. For example, in a study with RNA-seq and genotypes collected across multiple tissues, the expression of each tissue would be modeled independently[3,4]. More recent methods model multiple contexts simultaneously and leverage the sharing of genetic effects across contexts[8–10,12]. However, these approaches do not maximize predictive power because they ignore the intra-individual correlation of gene expression across contexts inherent to studies with repeated sampling, e.g., the Genotype-Tissue

---

Expression (GTEx) project[13] (Supplementary Fig. 1) or single-cell RNA-Sequencing (scRNA-Seq) experiments (Supplementary Fig. 2). Moreover, they build predictors which are mixtures of both context-specific and context-shared (pleiotropic) genetic effects, making it difficult to distinguish the relevant contexts for a disease gene, and are often computationally inefficient[9]. A recent approach by Wheeler et al.[14] does model correlated intra-individual noise with a linear-mixed model, but does not produce combined predictions of expression, reducing overall power. Finally, existing methods, with the goal of maximizing the number of discoveries made, may employ multiple testing strategies that either fail to control for all tests performed, (e.g., by controlling the false discovery rate (FDR) within each context separately[4,15]), or limit their discoveries as they are based on conservative FWER control (e.g., by using Bonferroni adjustment across all contexts[15]). Together, these shortcomings reduce power and interpretability of TWAS.

Here, we introduce CONTENT—CONtexT spEcific geNeTics—a method that leverages the correlation structure of multi-context studies to efficiently and powerfully generate genetic predictors of gene expression. Briefly, CONTENT decomposes the gene expression of each individual across contexts into context-shared and context-specific components[16], builds genetic predictors for each component separately, and creates a final predictor using both components. To identify genes with significant disease associations, CONTENT employs a hierarchical testing procedure (termed "hFDR"; see Supplementary Fig. 3)[17,18]. CONTENT has several advantages over existing methods. First, it explicitly accounts for intra-individual correlation across contexts, boosting prediction performance. Second, by building specific and shared predictors, it can distinguish context-shared from context-specific genetic components of gene expression and disease. Third, it employs a recently developed hierarchical testing procedure[18] to not only adequately control the FDR across and within contexts, but boost power in cases where a gene has a significant association to disease in multiple contexts. Fourth, this adjustment procedure allows for inclusion of other TWAS predictors[3,4,8–10,12], enabling approaches to be complementary in discovering associations. Finally, CONTENT is orders of magnitude more computationally efficient than several previous approaches.

We evaluate the performance of CONTENT over simulated data sets, GTEx[2,11,13], and a single-cell RNA-Seq data set[19–21]. We show in simulations that CONTENT captures a greater proportion of the heritable component of expression than previous methods (at minimum over 22% more), and that CONTENT successfully distinguishes the specific and shared components of genetic variability on expression. In applications to GTEx, CONTENT improves over previous context-by-context methods both in the number of genes with a significant heritable component (average 42% increase in significant gene-tissue pairs discovered) as well as the proportion of variability explained by the heritable component (average increase of 28%)[3,4]. Consistent with complex cell-type heterogeneity within tissues[22–25], we find that in applications to the single-cell data, genetic predictors at the cell-type level have substantially more context-specific heritability than the tissue-level models. We perform TWAS across 22 phenotypes using weights trained on GTEx and scRNA and find that CONTENT discovers over 51% additional independent, significantly associated loci relative to previous approaches. We provide CONTENT gene expression weights for both GTEx and the single-cell dataset at the TWAS/FUSION repository (http://gusevlab.org/projects/fusion/).

## Results
### Methods overview
We developed CONTENT, a method for generating genetic predictors of gene expression across contexts for use in downstream applications such as TWAS. Briefly, for each individual, CONTENT leverages our recently developed FastGxC method[16] to decompose the gene expression across $C$ contexts into one context-shared component and $C$ context-specific components (Fig. 1). Next, CONTENT builds genetic predictors for the shared component and each of the $C$ context-specific components of expression using penalized regression. We refer to these predictors as the CONTENT(Shared) and CONTENT(Specific) models. In addition, CONTENT generates genetic predictors of the total expression in each context by combining the context-shared and context-specific genetic predictors with linear regression. We refer to these predictors as the CONTENT(Full) models. A given gene may have CONTENT(Specific), CONTENT(Shared), and/or CONTENT(Full) models depending on the architecture of genetic effects.

We residualized the expression of each gene in each context over their corresponding covariates (e.g. PEER factors, age, sex, batch information) prior to decomposing and then fitting an elastic net with double ten-fold cross-validation for both CONTENT(Shared) and CONTENT(Specific). We examined the number of significantly predicted genes as well as the prediction accuracy (in terms of adjusted $R^2$) between the cross-validation-predicted and true gene expression per gene-context pair. To properly control the FDR for each method across contexts and genes, we employed a hierarchical FDR correction[17,18] (Supplementary Fig. 3 and Methods). We note that groups of contexts may comprise additional sources of pleiotropy (e.g., in GTEx the group of brain tissues may have their own shared effects in addition to the overall tissue-shared effects). The decomposition of CONTENT is flexible and can account for both levels of pleiotropy among contexts (see Supplementary Methods).

### CONTENT is powerful and well-calibrated in simulated data
We evaluate the prediction accuracy of CONTENT in a series of simulations and compare its performance to the context-by-context approach[3,4], which builds predictors by fitting an elastic net in each context separately, as well as UTMOST[9], which builds predictors over all contexts simultaneously using a group LASSO penalty. Implicitly, we compare to the method from[14] which decomposes expression into orthogonal context-shared and context-specific components, as the CONTENT(Shared) and CONTENT(Specific) models are related to these components (See Methods). We omit comparison to other TWAS methods as many of them are built on the same framework as the context-by-context approach, or require external data, such as curated DNase I hypersensitivity measurements[8,10,12].

We used simulation parameters from GTEx, the largest multi-context eQTL study to-date, as a guideline. Specifically, we generated gene expression and genotype data such that context-specific genetic effects mostly lie on the same loci as context-shared eQTLs, and context-specific eQTLs without context-shared effects are rare[2,16]. Intuitively, this framework assumes that, most often, SNPs affect expression of a gene in all contexts, but to a different extent in each context (rather than, for example, acting as an eQTL in only a single context). We varied the proportion of contexts with context-specific heritability, the number of context-specific eQTLs without a context-shared effect, the number of causal SNPs, and the intra-individual residual correlation while keeping the number of genes (1000), contexts (20), cis-SNPs (500) and the proportion of context-shared and context-specific heritability constant (0.3 and 0.1 respectively).

Throughout our simulations, CONTENT significantly outperformed the context-by-context and UTMOST approaches in terms of prediction accuracy of the total genetic contribution to expression variability (Fig. 2A, Supplementary Fig. 4). The average increase in adjusted $R^2$ between the true genetic component of expression and the CONTENT(Full) predictor was 0.22 over UTMOST ($p < 2\mathrm{e}{-16}$ paired two-way $t$-test) and 0.48 over the context-by-context approach ($p < 2\mathrm{e}{-16}$ paired two-way $t$-test). Across nearly the entirety of parameter settings, CONTENT generated context-specific components that were uncorrelated with the true context-shared components (mean

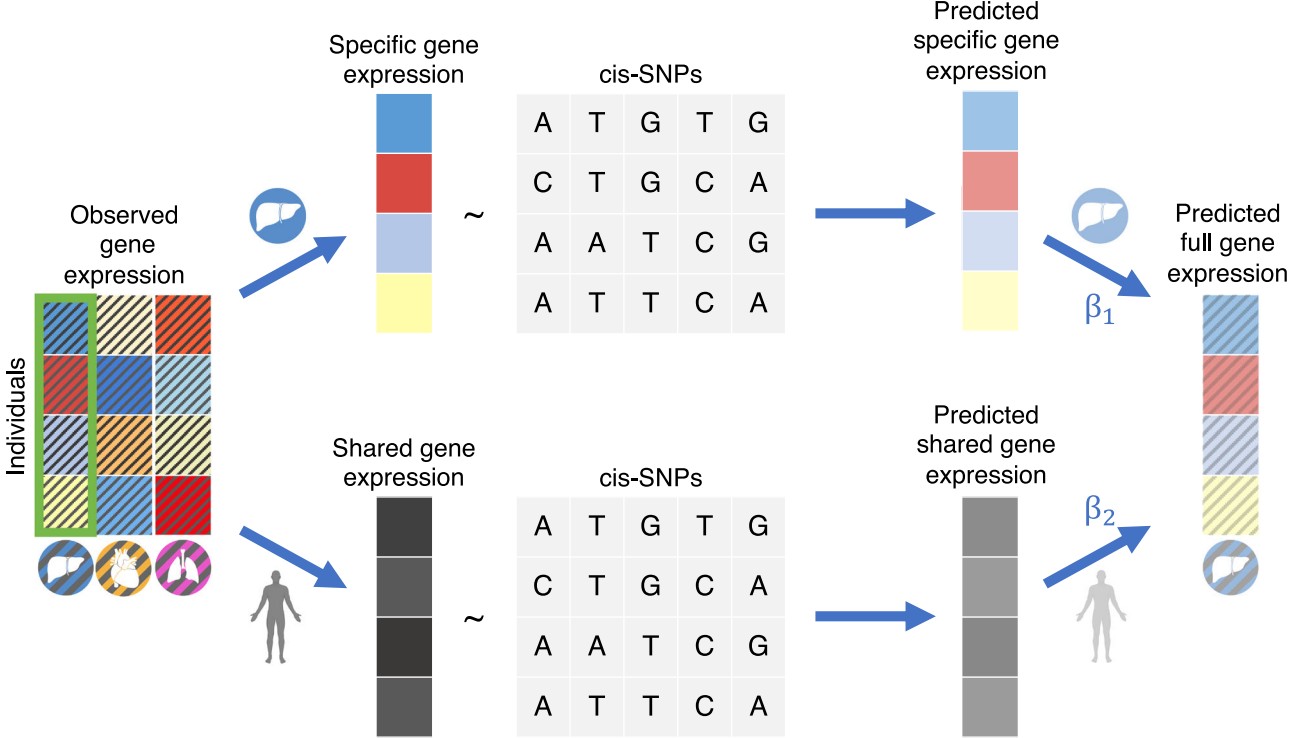

**Fig. 1 | An overview of the CONTENT approach.** CONTENT first decomposes the observed expression for each individual into context-specific and context-shared components following[16]. Then, CONTENT fits predictors for the context-shared component of expression as well as each context-specific component of expression (e.g., liver). Finally, for a given context, CONTENT combines the genetically predicted components into the full model using a simple regression. Icons were created with BioRender.com.

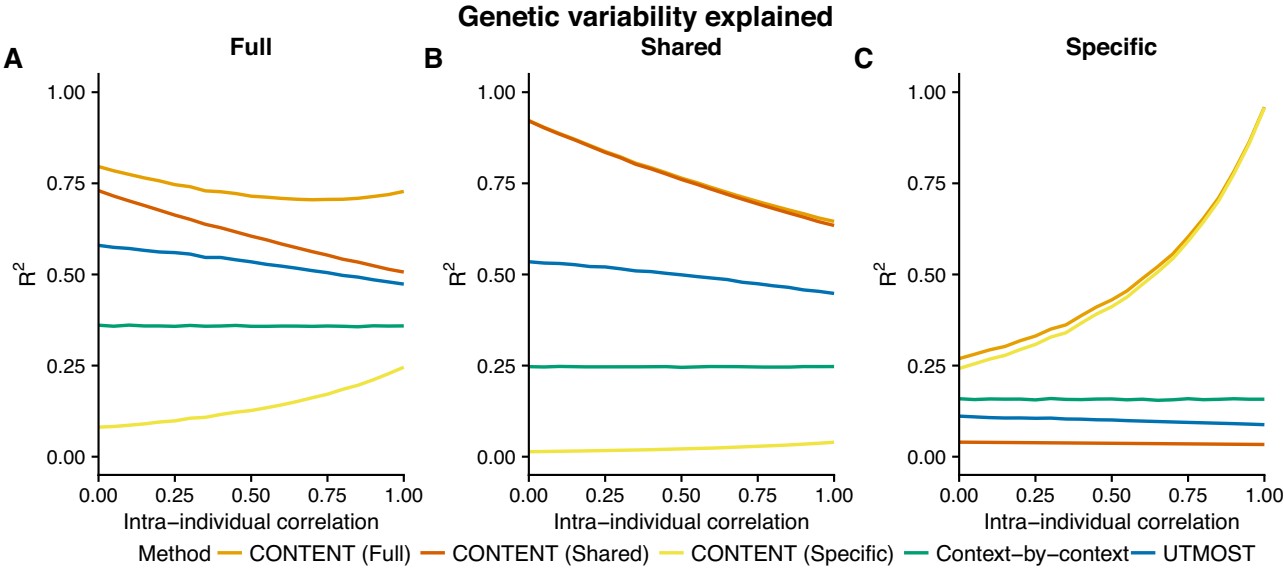

**Fig. 2 | CONTENT is powerful and well-calibrated in simulated data.** Accuracy of each method to predict the genetically regulated gene expression of each gene-context pair for different correlations of intra-individual noise across contexts. Mean adjusted $R^2$ across contexts between the true (**A**) full (context-specific + context-shared), **B** shared, and (**C**) specific genetic components of expression and the predicted component for each method and for different levels of intra individual correlation. The context-by-context approach and UTMOST output only a single predictor, and we show the variability captured by this predictor for each component of expression. CONTENT, however, generates predictors for all three components of expression, and notably, CONTENT(Specific) and CONTENT(Shared) capture their intended component of expression without capturing the opposite (i.e., the predictor for CONTENT(Specific) is uncorrelated with the true shared component of expression and vice versa). We show here the accuracy for each component and method on gene-contexts with both context-shared and context-specific effects, but show in Supplementary Fig. 4 the accuracy for all gene-contexts pairs.

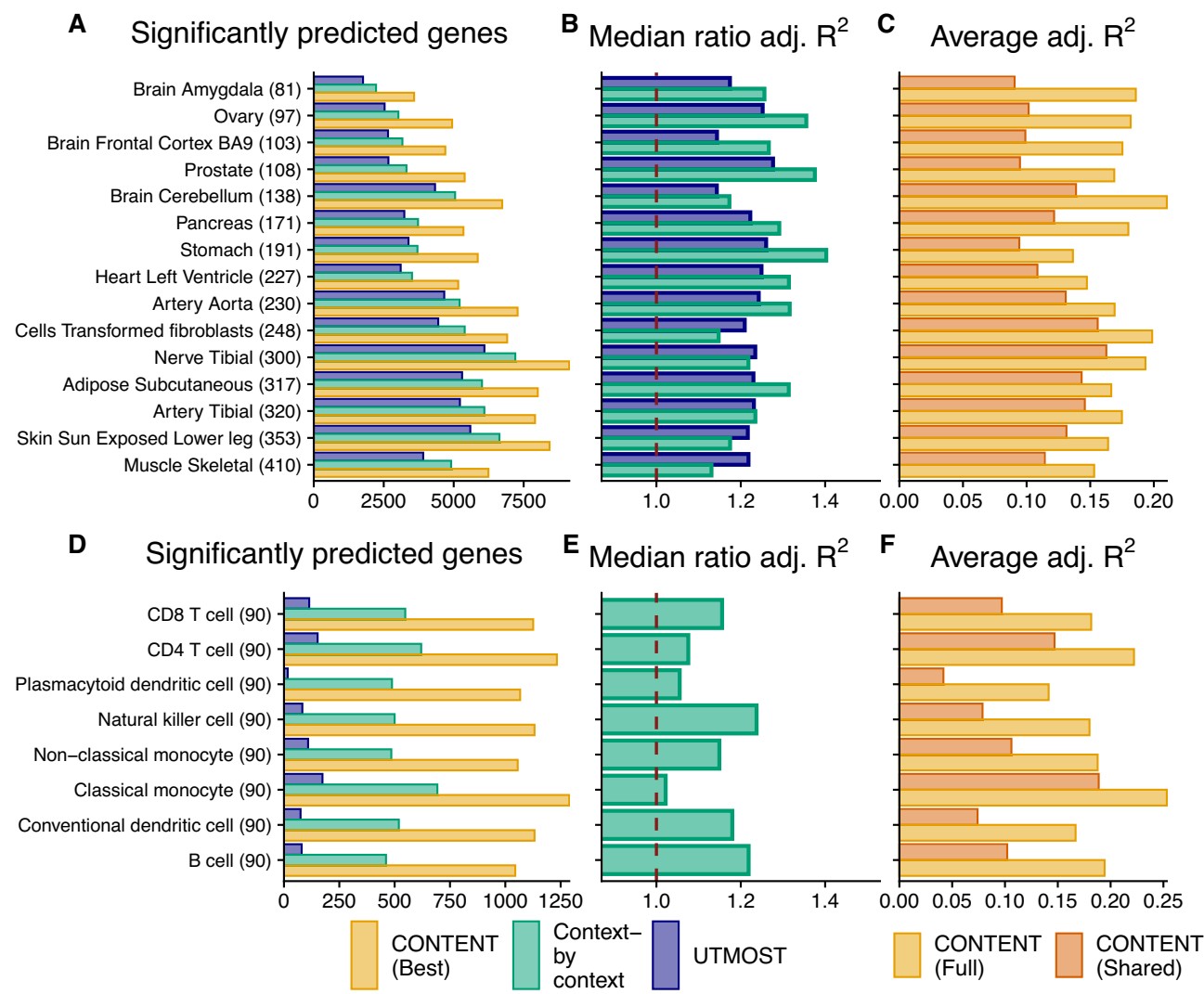

**Fig. 3 | CONTENT outperforms existing approaches in the GTEx and scRNA-seq CLUES datasets. A**, **D** Number of genes with a significantly predictable component (hFDR ≤ 5%) in GTEx (**A**) and CLUES (**D**); the sample sizes for each context are included in parentheses. **B**, **E** Ratio of expression prediction accuracy (adjusted $R^2$) of the best-performing cross-validated CONTENT model over the context-by-context (green) and UTMOST (blue) approaches (median across all genes significantly predicted by at least either method). Numbers above one indicate higher adjusted $R^2$ and thus prediction accuracy for CONTENT. **C**, **F** Prediction accuracy of CONTENT(Full) and CONTENT(Shared) when a gene-tissue has a significant shared, specific, and full model.

adjusted $R^2$ = 0.023, and vice versa 0.026; Fig. 2B, C). This property is central to the objective as it reduces confounding from pleiotropy in downstream applications such as context fine-mapping. As expected, the previous methods failed to disentangle the context-specific and context-shared components (Fig. 2B, C), since they were not developed with this property in mind. Our results were consistent under different values of the simulation parameters (Supplementary Figs. 5–8).

**CONTENT improves prediction accuracy over previous methods in the GTEx and CLUES datasets**

We next evaluated CONTENT, the context-by-context approach, and UTMOST in terms of prediction accuracy and power across 22,447 genes measured in 48 tissues of 519 European individuals in the bulk RNA-seq GTEx data set[2,11,13]. Due to computational issues (Supplementary Fig. 9), UTMOST was examined only on 22,307 genes rather than the entire data set of 22,447 genes. We show a comparison on this smaller set of genes in Supplementary Fig. 10. We also examined, for the first time in a large-scale TWAS context, a single-cell RNAseq data set from the California Lupus Epidemiology Study (CLUES)[19,20]. The

CLUES data set contained 9592 genes measured in 8 cell types in peripheral blood from 90 individuals.

In GTEx, CONTENT identified more gene-tissue pairs with a significantly predictable genetic component of expression (278,101 over 20,506 genes) than the context-by-context approach (195,607 over 17,723 genes) and UTMOST (167,865 over 11,442 genes) at an hFDR of 5% for all approaches. These results also held when using the traditional FDR approach within each context separately for all approaches (Supplementary Table 1 and Supplementary Fig. 11). We also compared the performance of each method on the union of genes that were significantly predicted (hFDR ≤ 5%) by at least one method. As CONTENT can generate up to three models (Shared, Specific, Full) for a given gene-tissue pair, and because each gene may have its own unique architecture (i.e. different proportions of specific or shared heritability), we selected the model that achieved the greatest cross-validated adjusted $R^2$. CONTENT greatly outperformed the context-by-context and UTMOST approaches across all tissues (average 28% and 22% increase in adjusted $R^2$ across tissues and genes; Fig. 3; Supplementary Figs. 10, 12). Further, for genes with significant CONTENT(Shared), CONTENT(Specific), and CONTENT(Full) predictors,

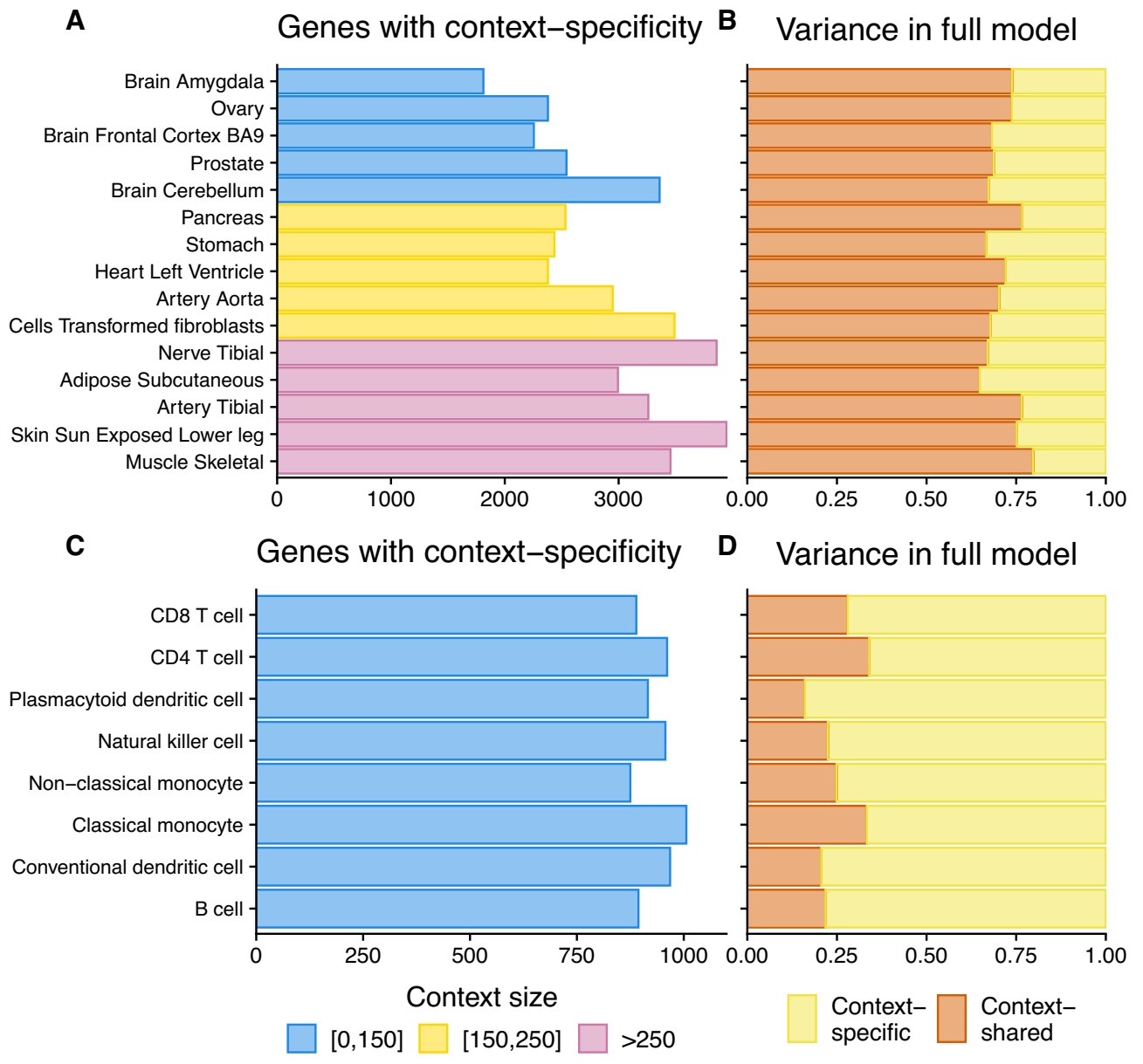

**Fig. 4 | Contribution of context-specific genetic regulation in GTEx and CLUES.**
**A**, **C** Number of genes with a significant (FDR ≤ 5%) CONTENT(Specific) model of
expression in GTEx (**A**) and CLUES (**C**). Color indicates sample size of context. **B**, **D** Proportion of expression variance of CONTENT(Full) explained by CON-
TENT(Specific) and CONTENT(Shared) for genes with a significant
CONTENT(Full) model.

prediction accuracy increases substantially with the addition of the
context-specific component to the context-shared component (aver-
age gain of CONTENT(Full) over CONTENT(Shared) adj. $R^2$ of 55.92%),
emphasizing the need to extend previous approaches[14] with CON-
TENT(Full) to build a powerful predictor.

Within the single-cell CLUES data set, CONTENT again out-
performed the context-by-context (in this case, cell type-by-cell type)
and UTMOST approaches, discovering 9080 heritable gene-cell type
pairs (5067 genes) whereas the context-by-context model and
UTMOST found 4314 (2355 genes) and 804 (288 genes) respectively.
The average improvement in adjusted $R^2$ of CONTENT over the
context-by-context model was 13.6%. In gene-cell type pairs with sig-
nificant CONTENT(Full), CONTENT(Specific), and CONTENT(Shared)
models, CONTENT(Full) improved the adjusted $R^2$ over CON-
TENT(Shared) by 104.09%. Once more, the improvement in variability
explained when including both the cell type-specific and cell type-

shared components highlights the need to consider both components
simultaneously when building a predictor.

**CONTENT discovers significant context-specific components of
expression in bulk multi-tissue and single-cell datasets**
Given the ability of CONTENT to disentangle context-shared and
context-specific variability, we examined the context-specific compo-
nents of expression in GTEx and CLUES. In GTEx, CONTENT discovered
128,985 gene-tissue pairs (19,765 genes) with a significant context-
specific genetic component of expression (Fig. 4; Supplementary
Fig. 13). As with previous reports[16,26], we found that testis was the tissue
with the greatest number of tissue-specific genetic components.
Nonetheless, we observe that the tissues with larger sample sizes more
frequently had significant context-specific components. Consistent
with previous works that have discovered extensive eQTL sharing
across tissues[2,26,27], we found that in gene-tissue pairs with a

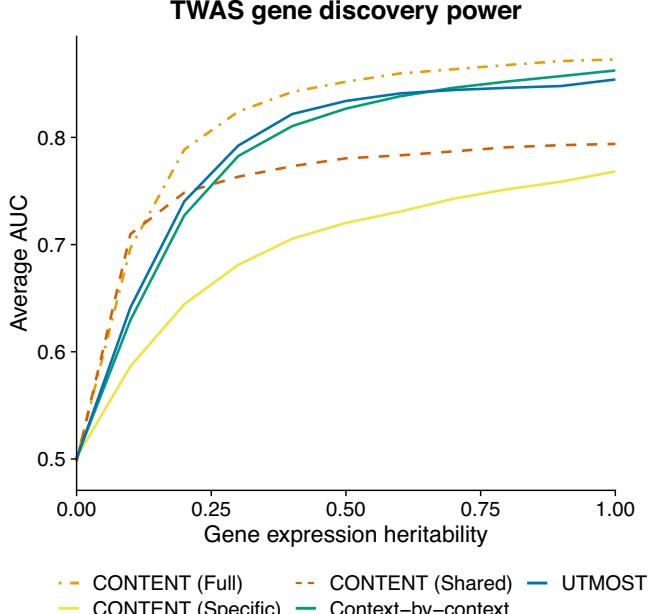

**TWAS gene discovery power**

**Fig. 5 | CONTENT(Full) is powerful, sensitive, and specific in simulated TWAS data.** Average AUC from 1000 TWAS simulations while varying the overall heritability of gene expression. Each phenotype (1000 per proportion of heritability) was generated from 300 (100 genes and 3 contexts each) randomly selected gene-context pairs' genetically regulated gene expression, and the 300 gene-context pairs' genetically regulated expression accounted for 20% of the variability in the phenotype. In genes with low heritability, CONTENT(Shared) performed similarly to CONTENT (Full), however CONTENT(Full) was the most powerful method in discovering the correct genes for TWAS across the range of heritability. CONTENT(Full) was significantly more powerful than UTMOST and the context-by-context approach at all levels of heritability.

CONTENT(Full) model, the variability explained was dominated by CONTENT(Shared) model—across tissues, the context-shared component explained on average 70% of the variability explained by CONTENT(Full).

In the CLUES data set, CONTENT discovered 7466 gene-cell type pairs (4658 genes) with a significant cell type-specific component of expression (hFDR ≤ 5%). We found that all cell types had a similar number of cell type-specific components, and emphasize that the sample size across all cell types was equivalent. In genes with a CONTENT(Full) model, the variability was often dominated by the cell type-specific variability (average 75% of the explained variability), unlike GTEx, in which the average tissue-specific variability explained only 30% of total variance. Consequently, we found that within the 20,433 genes in GTEx with any genetic component, 51.50% (10,522) had a significant shared component, whereas of the 5067 genes in CLUES with a genetic component, only 14.25% (722) had a shared component. This is consistent with complex celltype heterogeneity in bulk tissues[28] since there is more power to discover eQTLs with pleiotropy across the underlying cell types.

**CONTENT more accurately distinguishes disease-relevant genes than traditional TWAS approaches in simulated data**
We performed a simulation study in which we evaluated the sensitivity, specificity, and power of CONTENT, UTMOST, and context-by-context to implicate the correct gene in TWAS. In our experiments, we simulated a phenotype in which 20% of the variability was composed of the genetically regulated expression of 300 randomly selected gene-context pairs (100 genes and 3 contexts each). We simulated gene expression for 1000 genes across 20 contexts as before, however, to capture a range of

genetic architectures in the simulation, for each gene, we sampled from a standard uniform distribution to determine the proportion of shared variability. We varied the heritability of gene expression and considered power as a method's ability to discover the correct genes associated with a phenotype. To compare power, we calculated the area under receiver-operating curve (AUC) using the maximum association statistic for a given gene across contexts.

Across simulations, CONTENT(Full) was the highest powered in terms of gene discovery (Fig. 5). CONTENT(Shared) performed very similarly to CONTENT(Full) in the setting with the lowest heritability, however, our simulations show the necessity for CONTENT(Full) as it substantially outperforms both CONTENT(Specific) and CONTENT(Shared) across a range of heritabilities. Moreover, CONTENT(Full) significantly outperformed both the context-by-context approach and UTMOST. Specifically, the range of percent change in AUC of CONTENT(Full) over previous methods was as follows: CONTENT(Shared) 1.9–9.9%; CONTENT(Specific) 13.6–22.4%; UTMOST 2.2–8.6%; context-by-context 1.2–10.6%. Generally, we observed that CONTENT(Full) was its most powerful for genes in which there was both shared and specific effects, UTMOST was its most powerful in settings with high sharing, and the context-by-context approach was its most powerful in settings with low sharing and high specificity of genetic effects within contexts.

**Application of CONTENT to TWAS yields additional discoveries over previous methods**
We performed TWAS across 22 complex traits and diseases collected from a variety of GWAS[29–42] using weights trained by CONTENT, UTMOST and the context-by-context approach on GTEx and CLUES. We passed forward weights to FUSION-TWAS[3]—a software that performs TWAS using GWAS summary statistics, user-specified gene expression weights, and an LD reference panel—for a gene-context pair if the pair's expression was predicted at a nominal p-value less than 0.1 (See Methods; Supplementary Fig. 16).

Across all traits at an hFDR of 5%, CONTENT discovered a median of 51% (range of 5–178%) and 135% (51–400%) more associations (unique TWAS loci) than the context-by-context approach and UTMOST respectively with GTEx weights, and 62% (0–289%) and 101% (47–600%) more loci than the context-by-context approach and UTMOST respectively with weights built from the CLUES dataset (Table 1). We find that, with GTEx weights, the associations implicated by the context-by-context approach had more overlap with the associations implicated by CONTENT(Specific) (median Jaccard similarity (JS) across traits = 0.419) than CONTENT(Shared) (JS = 0.234). This is consistent with our simulation results in which the context-by-context approach was most powerful in cases of high context-specificity and low context-sharing (Supplementary Figs. 14, 15). Conversely, the associations discovered by UTMOST, which leverages pleiotropy, had slightly higher overlap with CONTENT(Shared) (JS = 0.221) than CONTENT(Specific) (JS = 0.177). With CLUES weights, the context-by-context approach again had greater similarity with CONTENT(Specific) (JS = 0.291) than CONTENT(Shared) (JS = 0.098), however UTMOST discovered TWAS genes that had similar overlap between CONTENT(Shared) (JS = 0.119) and CONTENT(Specific) (JS = 0.135). As UTMOST, CONTENT, and the context-by-context approach discovered both overlapping and unique associations, we suggest that the approaches complement—rather than replace—one another.

We next compared the different CONTENT models to understand their properties in real data. With GTEx weights, CONTENT(Full) replicated an average of 98.3% and 67.3% of the associations discovered by CONTENT(Shared) and CONTENT(Specific) respectively (hFDR ≤ 5%). CONTENT(Full) replicated an average of 81.2% and 61.6% of the associations discovered by CONTENT(Shared) and

**Table 1 | CONTENT outperforms existing methods in TWAS across 22 complex traits and diseases**

| Trait | GTEx | | | | | | CLUES | | | | | |
|---|---|---|---|---|---|---|---|---|---|---|---|---|
| | Context-by-context | UTMOST | CONTENT (All) | CONTENT (Full) | CONTENT (Specific) | CONTENT (Shared) | Context-by-context | UTMOST | CONTENT (All) | CONTENT (Full) | CONTENT (Specific) | CONTENT (Shared) |
| AD | 17 | 9 | **24** | 20 | 20 | 11 | 7 | 5 | **15** | 9 | 13 | 3 |
| Asthma | 155 | 90 | **237** | 181 | 195 | 67 | 74 | 63 | **127** | 101 | 104 | 34 |
| Bipolar | 42 | 45 | **83** | 63 | 65 | 39 | 9 | 14 | **35** | 20 | 25 | 5 |
| CAD | 10 | 11 | **23** | 18 | 15 | 8 | 6 | 6 | **10** | 7 | 6 | 0 |
| CKD | 26 | 19 | **42** | 31 | 32 | 18 | 2 | 4 | **6** | 5 | 5 | 1 |
| Crohn's | 77 | 63 | **95** | 73 | 83 | 47 | 27 | 22 | **44** | 30 | 37 | 9 |
| Eczema | 32 | 13 | **57** | 44 | 41 | 10 | 8 | 5 | **11** | 9 | 9 | 3 |
| FastGlu | 16 | 8 | **19** | 12 | 14 | 7 | 3 | 3 | 6 | 6 | 6 | 0 |
| HDL | 58 | 29 | **79** | 60 | 73 | 36 | 21 | 14 | **28** | 23 | 25 | 6 |
| IBS | 9 | 5 | **25** | 20 | 16 | 3 | 3 | 1 | **7** | 5 | 6 | 1 |
| LDL | 89 | 57 | **132** | 107 | 116 | 58 | 47 | 29 | **51** | 40 | 44 | 14 |
| Lupus | 93 | 54 | **129** | 94 | 104 | 51 | 36 | 27 | **58** | 42 | 48 | 11 |
| MDD | 99 | 79 | **169** | 132 | 134 | 62 | 20 | 29 | **47** | 32 | 39 | 3 |
| MS | 20 | 10 | **42** | 32 | 25 | 9 | 9 | 7 | **11** | 8 | 10 | 5 |
| PBC | 62 | 42 | **65** | 55 | 58 | 33 | 21 | 14 | **30** | 24 | 26 | 6 |
| Psoriasis | 47 | 22 | **58** | 46 | 41 | 16 | 13 | 10 | **21** | 17 | 16 | 6 |
| RA | 73 | 56 | **99** | 79 | 86 | 46 | 40 | 20 | **51** | 33 | 45 | 9 |
| Sarcoidosis | 19 | 13 | **30** | 27 | 26 | 8 | 6 | 4 | 6 | 6 | 4 | 2 |
| Sjogren | 17 | 9 | **31** | 25 | 21 | 6 | 4 | 2 | **7** | 6 | 6 | 1 |
| T1D | 77 | 64 | **109** | 88 | 84 | 49 | 26 | 23 | **41** | 36 | 29 | 13 |
| T2D | 193 | 115 | **246** | 208 | 205 | 112 | 76 | 76 | **112** | 77 | 98 | 17 |
| Ulc colitis | 16 | 10 | **40** | 30 | 26 | 7 | 5 | 4 | **11** | 9 | 7 | 2 |

TWAS results (unique loci, merging genes within 1MB) across 22 complex traits and diseases using weights output by CONTENT, UTMOST, and the context-by-context method. CONTENT(All) refers to the collection of all loci output by at least one CONTENT model. CONTENT(Full) added an average of 15% and 19% of gene-trait discoveries over the CONTENT(Shared) and CONTENT(Specific) approaches together at an hFDR of 5% in GTEx and CLUES respectively. Numbers in bold font indicate the method with the greatest number of discoveries. AD Alzheimer's disease, CAD Coronary Artery Disease, CKD Chronic Kidney Disease, Crohn's Crohn's Disease, FastGlu Fasting Glucose, GFR Glomerular filtration rate, HDL High-density lipoprotein, IBS Irritable bowel syndrome, LDL Low-density lipoprotein, Lupus Systemic lupus erythematosus, MDD Major depressive disorder, MS Multiple sclerosis, PBC Primary biliary cholangitis, RA Rheumatoid arthritis, Sjogren Sjögren's syndrome, T1D Type 1 diabetes, T2D Type 2 diabetes, TG Triglycerides, Ulc colitis Ulcerative colitis.
See Supplementary Table 2 for GWAS trait information

CONTENT(Specific) respectively with the CLUES weights. Notably, CONTENT(Full) is the best predictor out of all the CONTENT models on average, particularly when there exist both shared and specific effects. Consequently, across all traits, the inclusion of CONTENT(Full) with CONTENT(Shared) and CONTENT(Specific) led to an average increase of 12% and 21% in the number of genes with significant TWAS associations with GTEx weights and CLUES weights respectively.

We investigated the genes implicated by CONTENT(Full) that were not significant in CONTENT(Shared) or CONTENT(Specific) and found that many of the discoveries replicated known gene-trait associations. For example, CONTENT(Full) discovered a significant association of fasting glucose levels and CAMK2 ($p$ = 2.44e-23, brain cortex), a gene responsible for calcium signaling and regulation of hepatic glucose production[43], as well as BLVRA (4.21e-06, CD8 T cell), a gene involved in insulin signaling and likely metabolic syndrome[44]. Furthermore, CCL2, which is thought to be involved in HDL internalization and cholesterol efflux[45], was not implicated by either CONTENT(Shared) or CONTENT(Specific), but was implicated in the TWAS of HDL with CONTENT(Full) ($p$ = 2.30e-08, small intestine terminal ileum). CONTENT(Full) also discovered a significant association of F2 (prothrombin) and primary biliary cirrhosis (PBC) (1.47e-07, liver), whereas CONTENT(Shared) and CONTENT(Specific) did not; PBC patients have been shown to have higher prothrombin times than controls[46]. Moreover, CONTENT(Full) discovered an association of GIT1−a gene involved with synaptic transmission and plasticity[47,48]−with bipolar disorder (BIP; B cell, $p$ = 3.20e-06) as well as an association of GSDMB−a gene involved with airway remodeling and airway-hyperresponsiveness[49]−and asthma (CD4 T cell, $p$ = 1.25e-20).

Moreover, the genes implicated by CONTENT but neither UTMOST nor the context-by-context approach (at an hFDR of 5%) replicated previously associated gene-trait pairs, several of which with known biological relationships to the trait of interest. Within Alzheimer's disease, these genes included VGF[50], FZD4[51], and TRPV6 (a transient receptor potential channel)[52,53] with the GTEx weights, as well as IRF7[54] and GANC[55] with CLUES weights. Additionally, in Crohn's disease, CONTENT implicated the following genes, whereas previous methods did not: STAT3[56] and CTBP2[57] with GTEx weights, as well as ATG16L[58] and PKAR2A[59] using CLUES weights. For major depression disorder (MDD), CONTENT implicated SYN2M[60] and CYB56AD1[61] using GTEx weights, and GAB1[62], TLR4[61] and ARL3[63] using CLUES weights.

As the individuals comprising the GTEx and CLUES datasets are disjoint, we also investigated whether using both datasets could highlight relevant biological genes (akin to a replication study). We first examined LDL genes and found SORT1, which alters plasma LDL levels (GTEx min. $p$ = 2.15e-251, CLUES min. $p$ = 2.41e-19)[64–66]. We next found an association between S100A4, S100A8, S100A10, S100A11 as well as S100A12 (part of the epidermal differentiation complex) and Eczema using both datasets (S100A10 $p$ = 2.78e-41, $p$ = 2.90e-11)[67,68]. Additionally, when we looked at discoveries made with GTEx and CLUES weights for Alzheimer's disease, we found MARK4 ($p$ = 8.72e-20, $p$ = 6.39e-63), a gene associated with tau phosphyrlation in granulo-vacuolar degeneration bodies[69]. Finally, both sets of weights produced a significant association of immune checkpoint gene CTLA4 ($p$ = 1.71e-11, $p$ = 2.28e-21) with Rheumatoid Arthritis[70].

While CONTENT discovered substantially more loci and genes than previous approaches, we also wished to verify that it does not enrich for false positives. To do so, we performed an analysis similar to one carried out by Ndungu et al.[71]. Briefly, Ndungu et al. evaluated the extent to which TWAS associations may be driven by horizontal pleiotropy or linkage disequilibrium by examining TWAS associations for a set of genes with a known causal relationship to a set of metabolites. In our analyses, we examined the within-locus (±1 Mb) rank of

the causal TWAS gene with its suspected metabolite when using weights built by CONTENT and the context-by-context approach on the GTEx dataset. To order genes within a method, we first filtered for statistically significant gene-context-metabolite associations, then sorted genes by their maximum absolute TWAS association statistic between a given metabolite across contexts (and models for CONTENT). In line with our applications of TWAS to GTEx and CLUES, CONTENT discovered additional loci that were not discovered by the context-by-context approach (39 compared to 36 of 58 known gene-metabolite pairs; Supplementary Table 3). Moreover, despite having more models built per locus, CONTENT ranked the known causal gene similarly to the context-by-context approach on the intersection of gene-metabolite pairs discovered by both methods (CONTENT average rank of 2.257 compared to context-by-context rank of 2.371, where a ranking of 1 is ideal).

## Discussion

In this work, we introduce CONTENT, a computationally efficient and powerful method to estimate the genetic contribution to expression in multi-context studies. CONTENT can distinguish the context-shared and context-specific components of genetic variability and can account for correlated intra-individual noise across contexts. Using a range of simulation and real studies, we showed that CONTENT outperforms previous methods in terms of prediction accuracy of the total genetic contribution to expression variability in each context. We also found that when there exists a gene with a genetic component of expression, the heritability is often dominated by the context-specific effects at the single-cell level, but at the tissue level, the heritability is dominated by the context-shared effects. Finally, CONTENT was more powerful, specific, and sensitive than previous approaches in applications to TWAS.

Using weights trained by CONTENT, UTMOST and the context-by-context approach, we discovered 12,150 unique gene-trait associations through TWAS. To our knowledge, we present the first application of TWAS trained on a single-cell RNAseq dataset for a collection of 90 individuals' PBMCs. For both the weights generated by GTEx and CLUES, CONTENT was largely more powerful than UTMOST and the context-by-context approach in TWAS. However, we emphasize that the approaches often capture genes unique to each approach. Each method may therefore complement each other and may be combined in TWAS to maximize the number of discoveries made as different methods are likely favorable under different genetic architectures. Though we show that CONTENT may be useful in fine-mapping the specific tissue relevant for a TWAS association in simulations, we note that fine-mapping to the correct tissue in real data is a particularly difficult task. For example, throughout this manuscript, we assume that the causal tissue is included in the measured tissues, however, when this is not the case, CONTENT and all TWAS approaches may associate an incorrect, correlated tissue. For example, in the case of chronic kidney disease, CONTENT implicated GATM−a gene thought to be involved with kidney disease and GFR levels[72–74]−however, the significant association was within the thyroid. This may be due to the fact that kidney expression is not measured in this version of the GTEx dataset. Future work may explore using the CONTENT-trained weights and jointly fitting all TWAS Z scores, or otherwise accounting for missingness.

We also leveraged recently developed methodology for controlling the false discovery rate when summarizing significantly predicted genes, gene-contexts, and TWAS associations[17,18]. This approach has been shown to effectively control the FDR across contexts in eQTL studies, and to our knowledge, it is the first time such an approach has been used to effectively control the FDR when predicting expression values and when making discoveries using TWAS. While our analyses focused on the comparison of CONTENT, UTMOST, and the

context-by-context approach, we emphasize that by using this type of false discovery correction, all methods can be used in combination with one another, rather than in replacement of one another. For downstream analyses, combining all prediction methods is crucial, as certain genes or gene-context pairs may be (better) predicted by one method and not others. In the GTEx data for example, when we included models built by UTMOST and the context-by-context approach to the correction scheme for CONTENT, the number of genes for which there was a significant model for a given tissue increased on average by 7.56%.

Importantly, neither UTMOST nor the context-by-context method distinguishes the context-specific and context-shared components of genetic effects on expression. Implicitly, by modeling all contexts independently, the context-by-context fit is best-suited for cases in which there is no effect-sharing across contexts (Supplementary Fig. 12). As UTMOST considers all contexts simultaneously, its power is maximized in cases where the genetic effects are mostly shared (Supplementary Fig. 12). Additionally, these methods do not account for the shared correlated residuals between samples, thus they do not maximize their predictive power.

While a previous approach proposed by Wheeler et al.[14] does model the correlated intra-individual noise, CONTENT offers several advantages. The previous decomposition does not include an option to leverage both the context-shared and context-specific components of expression to form a final predictor of the observed expression for a given context. We show that this is especially crucial in the context of single-cell data wherein the prediction accuracy for a given gene-context increases drastically when using both components (Fig. 3). Further, without properly combining both components (e.g. via regression), the context-specific genotype-expression weights produced by the previous decomposition may have the incorrect sign, as they are considered residuals of the context-shared component and are not properly re-calibrated to the observed expression. Unlike the decomposition proposed by CONTENT, this previous approach also does not intuitively allow for additional sources of pleiotropy or effects-sharing (see Supplementary Information for discussion of brain-level sharing in GTEx; Supplementary Figs. 17–20 and Table 4). Finally, the decomposition used in the previous method is based on a linear mixed model fit on a per-gene basis, and is therefore less computationally efficient.

Notably, a limitation of TWAS methods in general is interpretability, as associations may be confounded by linkage disequilibrium or horizontal pleiotropy[71,75]. We emphasize that CONTENT discovered substantially more independent loci than previous methods, however, since CONTENT is more powerful than previous methods, it may build more models within a given locus relative to previous approaches. We performed a brief set of analyses in line with Ndungu et al.[71], in which we evaluated the ability of TWAS approaches to associate the suspected causal gene to a collection of metabolites. Despite CONTENT building more models than the context-by-context approach, it prioritized suspected genes the same as or better than the context-by-context approach in addition to discovering several more loci than did the context-by-context approach (Supplementary Table 3). We therefore conclude that, similarly to GWAS fine-mapping studies, resolution of downstream TWAS fine-mapping methods (e.g., FOCUS[75]) should increase with the use of our models, as our gain in performance is akin to that expected from an increase in sample size. Moreover, since CONTENT discovers additional loci over previous approaches, it undoubtedly will present additional useful information for such studies.

In this manuscript we focused on prediction of the total genetic contribution to expression as well as the context-shared and context-specific components of expression. Nonetheless, future work using the methodology presented here can be extended to a wide variety of problems. Primarily, the decomposition can be used to efficiently estimate Gene × Context heritability using existing software for heritability estimation, e.g., GCTA[76], on the decomposed components offering computational speed up over existing methods for cross-context heritability estimation[27]. Additionally, the decomposed components from CONTENT may also be included in previous approaches, e.g. UTMOST, to gain further power. Further, by training each method on the single-cell level data, we offer researchers the means to pursue their own association analyses at a lower level of granularity than was previously available.

Notably, we found that single-cell data may have lower levels of effects-sharing than tissue-level data. While this may be due to genuine biological differences in genetic regulation, this finding is also consistent with a large degree of sharing of cell types across contexts. For example, endothelial cells can be found in tissues such as breast, endometrium, esophagus, eye, heart muscle, liver, lung, ovary, pancreas, placenta, prostate, skeletal muscle, and skin and often make up a substantial fraction of the collected tissue[77,78]. We believe our work is consistent with this observation: Primarily, the proportion of eGenes (genes with a heritable component of expression) that also have a shared component is substantially lower at the single cell level compared to the proportion at the bulk, tissue level. What's more is that the ability to discover context-specific components of expression is indeed related to sample size in the GTEx dataset. Despite the above, and having a lower number of individuals in the single-cell data, we discover a greater proportion of genes with a context-specific component than in GTEx. Further, when there exists a CONTENT(Full) model, it is dominated by the specific variability at the single-cell level, whereas it is dominated by the shared variability at the tissue level. Nonetheless, as this finding, to our knowledge, was previously unappreciated, it warrants further investigation.

In summary, we present an approach for generating context-shared and context-specific predictors that is much simpler than previous approaches[14,16]. Moreover, unlike previous methods, we offer a way to combine both predictors, as well as extend the decomposition to additional levels of pleiotropy. Finally, we show utility of existing hierarchical FDR correction methods to properly adjust for analyses that take advantage of multiple methods as well as investigate genes in the space of multiple contexts. The increased prediction accuracy, specificity, computational speed, and hierarchical testing framework of CONTENT will be paramount to unveiling context-specific effects on disease as well as uncovering the mechanisms of context-specific genetic regulation.

## Methods

### An overview of the CONTENT model

In this section, we detail the assumed generative model and objectives of CONTENT. CONTENT is based on the methodology and decomposition of a previous work by Lu et al. FastGxC[16]. In brief, like FastGxC, we assume that the expression of an individual in a given gene and context is a combination of a context-shared genetic component that is shared across different contexts and a context-specific genetic component that is specific to a context, that is

$$E_c = E_G^{\text{Shared}} + E_{G,c}^{\text{Specific}} + \varepsilon_c \tag{1}$$

$$E_G^{\text{Shared}} = \mathbf{g}\boldsymbol{\beta} \tag{2}$$

$$E_{G,c}^{\text{Specific}} = \mathbf{g}\boldsymbol{\gamma}_c \tag{3}$$

where $E_c$ denotes the expression of the individual at the gene in context $c$, $E_G^{\text{Shared}}$ and $E_{G,c}^{\text{Shared}}$ denote the components of the expression due to context-shared and context-specific genetic effects respectively, $\boldsymbol{\beta}$

and $\gamma_c$ represent the context-shared and context-specific cis-genetic effects respectively, $g$ the individual's cis-genotypes and $\varepsilon_c \sim N(0, \sigma_c^2)$ represents the environmental effects (and non-cis-genetic effects) on the individual's gene expression.

The objective of CONTENT is to build a genetic predictor of context-specific phenotypes. While previous work has focused on building powerful genetic models for $E_c$, we aim to build unbiased models that partition and estimate the context-shared $g\beta$ and context-specific terms $g\gamma_c$. Specifically, we aim to maximize the power to detect the context-specific terms, allowing some leniency in the accuracy of context-shared terms, as we are interested in context-specific effects. Moreover, as a context-specific predictor can be used in downstream analyses to identify the specific context(s) through which genetic variation manifests its effect on the phenotype and disease risk, we also aim to minimize the correlation between the predicted context-specific component and the true context-shared component. Finally, our method must account for the correlated intra-individual noise across contexts, and do so in a computationally efficient manner.

## Decomposing multilevel data

Many genomic datasets, such as those of GTEx, have a multilevel nature; first the individuals are sampled, and second an individual is measured in each context. To take the multilevel structure of the data into account, the observed expression on gene $j$ can be decomposed into an offset term, a between-individual component and a within-individual component[79]. That is, if $E_{ijc}$ denotes the observed expression level for individual $i$ ($i = 1, \ldots, I$) on gene $j$ ($j = 1, \ldots, J$) and context $c$ ($c = 1, \ldots, C$), $E_{ijc}$ can be decomposed as

$$E_{ijc} = E_{j.} + (E_{ij.} - E_{j.}) + (E_{ijc} - E_{ij.}) \tag{4}$$

where $E_{j.} = \frac{1}{I \times C} \sum_{i=1}^{I} \sum_{c=1}^{C} E_{ijc}$ the mean expression of gene $j$ computed over all ($I$) individuals and all ($C$) contexts, and $E_{ij.} = \frac{1}{C} \sum_{c=1}^{C} E_{ijc}$ the mean expression of individual $i$ on gene $j$, computed over all contexts. In (4), $E_{j.}$ is a term that is constant across individuals and contexts for each gene, $(E_{ij.} - E_{j.})$ is the between-individuals deviation, and $(E_{ijc} - E_{ij.})$ is the within-individual deviation of the expression on gene $j$ in context $c$.

Variables that differ between but not within individuals, e.g. sex and genotype, will have an effect on $(E_{ij.} - E_{j.})$ but not on $(E_{ijc} - E_{ij.})$. On the other hand, variables that change within individuals but are the same between individuals, e.g. the genetic effect on a specific context, will have an effect on $(E_{ijc} - E_{ij.})$ but not on $(E_{ij.} - E_{j.})$.

In the context of estimation, we first center and scale the expression of gene $j$ in each context $c$, i.e., $\frac{1}{I} \sum_{i=1}^{I} E_{ijc} = 0$ and $\frac{1}{I} \sum_{i=1}^{I} E_{ijc}^2 = 1$. Therefore, $E_{j.} = \frac{1}{I \times C} \sum_{i=1}^{I} \sum_{c=1}^{C} E_{ijc} = 0$, and equation (4) simplifies to:

$$E_{ijc} = \underbrace{E_{ij.}}_{E_{ij}^{\text{Shared}}} + \underbrace{(E_{ijc} - E_{ij.})}_{E_{ijc}^{\text{Specific}}} \tag{5}$$

## A formal description of CONTENT

We use the simplified decomposition in equation (5) to build genetic predictors of context-specific effects while accounting for the correlated intra-individual noise across contexts. Intuitively, the between-individuals variability serves as the component of expression that is shared across contexts, $E^{\text{Shared}}$, and the deviance from this shared component (i.e. the within-individual variability) serves as the context-specific component of expression, $E^{\text{Specific}}$. Moreover, treating the context-specific component as a deviance from the context-shared component leads the decomposition to have the property that as the correlation of intra-individual noise across contexts increases, the

power to detect context-specificity also increases. In addition, the decomposition generates context-shared and context-specific components of expression that are orthogonal to each other. Further rationale for using the decomposed expression is included Supplementary Note 1 and the text by Lu et al.[16]. Lu et al. also include a description of the decomposition's equivalence to a linear mixed model.

For a single gene $j$, CONTENT takes as input centered, scaled, and residualized (over a set of covariates) expression measured across $I$ individuals in $C$ contexts and an $I \times m$ genotype matrix $G_j$ with $m$ measured cis-SNPs for gene $j$. CONTENT then decomposes the expression vectors into $C$ context-specific components and a single context-shared component by simply calculating the mean of expression for each individual across contexts, and setting the context-specific expression for context $c$ as the difference between the observed expression of context $c$ and the calculated context-shared (mean) expression. As it has been observed that cis-genetic effects may be sparse and that the elastic net may perform best relative to other penalized linear models in the context of genetically regulated gene-expression[4,14], CONTENT fits $C + 1$ penalized linear models for the $C + 1$ expression components using an elastic net. Lastly, CONTENT generates a final genetic predictor of expression by combining the context-shared and context-specific components. Importantly, as the context-specific component is a deviance from the context-shared component, the sign of the context-specific component must be properly realigned when combining both components of expression to make a final predictor. We refer to this linear combination of expression components as the "full" model of CONTENT and fit it using a simple linear regression:

1. Obtain $\mathbf{E}_j^{\text{Shared}}$ and $\mathbf{E}_{jc}^{\text{Specific}}$ from the decomposition across all individuals.
2. Generate cis-genetic predictors of each component using cross-validated elastic net:
   (a) Fit cross-validated elastic net regressions for the shared and specific components:

   $$\mathbf{E}_j^{\text{Shared}} = \boldsymbol{\alpha}^{\text{Shared}} + \mathbf{G}_j \boldsymbol{\beta} + \boldsymbol{\varepsilon}^{\text{Shared}} \tag{6}$$

   $$\mathbf{E}_{jc}^{\text{Specific}} = \boldsymbol{\alpha}_c^{\text{Specific}} + \mathbf{G}_j \boldsymbol{\gamma}_c + \boldsymbol{\varepsilon}_c^{\text{Specific}} \tag{7}$$

   (a) Use the estimates to generate genetic predictors of each component:

   $$\hat{\mathbf{E}}_{jG}^{\text{Shared}} = \hat{\boldsymbol{\alpha}}^{\text{Shared}} + \mathbf{G}_j \hat{\boldsymbol{\beta}} \tag{8}$$

   $$\hat{\mathbf{E}}_{jcG}^{\text{Specific}} = \hat{\boldsymbol{\alpha}}_c^{\text{Specific}} + \mathbf{G}_j \hat{\boldsymbol{\gamma}}_c \tag{9}$$

3. Regress the expression of context $c$ onto the context-shared and context-specific components:

   $$\mathbf{E}_{jc} = \boldsymbol{\alpha}_c^{\text{Full}} + \hat{\mathbf{E}}_{jG}^{\text{Sh.}} w_{jc}^{\text{Sh.}} + \hat{\mathbf{E}}_{jcG}^{\text{Sp.}} w_{jc}^{\text{Sp.}} + \boldsymbol{\varepsilon}_{jc} \tag{10}$$

Where "Sh." and "Sp." indicate "shared" and "specific" respectively, $\alpha$ represents the offset within each regression, and all $\varepsilon$ are assumed to be from a normal distribution with mean 0 and standard deviation that is a function of the given outcome.

We save for each gene the set of estimated regression weights $\hat{w}_{jc}^{\text{Shared}}$ and $\hat{w}_{jc}^{\text{Specific}}$ from equation (10) for use in downstream analyses. Namely, in TWAS, each context receives a single vector of weights, and to test the association of a gene-context's full model to a trait, we simply use a weighted sum of the predictors learned from equation (3),

$\hat{w}_{jc}^{\text{Sh.}} \cdot \hat{\boldsymbol{\beta}} + \hat{w}_{jc}^{\text{Sp.}} \cdot \hat{\boldsymbol{\gamma}}_c$. We also use the same procedure for the context-specific weight to ensure the correct directionality. To test for significance of genetic effects (i.e. to call an eGene or eAssociation), we correlate each component of expression—the context-shared, context-specific, and full—to its corresponding genetically predicted value. More concretely, we perform a likelihood ratio test with one degree of freedom for the specific and shared models, and a likelihood ratio test with two degrees of freedom for the full model (the null model for all tests contains no genetic predictors of expression).

### Controlling the false discovery rate across contexts

Generally, methods for building genetic predictors of expression or TWAS predictors leverage either Bonferroni correction or false discovery rate (FDR). Nonetheless, using a Bonferroni correction may be too stringent (for example, as tests across contexts may be correlated), and using FDR within each context or across all contexts simultaneously may lead to an inflation or deflation to the false disovery proportion within certain contexts[17]. To simultaneously control the FDR across all contexts at once, a hierarchical false discovery correction—treeQTL—was developed[17]. The treeQTL procedure leverages the hierarchical structure of a collection of tests (e.g. gene level and gene-context level) to properly control the FDR across an arbitrary number of contexts and levels in the hierarchy as well as boost power in cases where a gene has a significant association in multiple contexts[6,17,18]. (See Supplementary Methods for further intuition.)

Notably, using CONTENT, our testing hierarchy contains 3 levels; (1) at the level of the gene, (2) at the level of the context, and (3) at the level of the method or model (Supplementary Fig. 3). Intuitively, a gene may contain a genetic component that is shared across all contexts, or a given context may have its own genetic architecture. In CONTENT, a given context may have its own genetic predictor from either the context-specific component or the full model. Using treeQTL with this structure is robust across multiple contexts, and since the tree is structured such that a specific method/model is at the final level of testing for a context, it enables incorporation of additional models trained from other approaches (such as those fit on a context-by-context basis or by UTMOST). Moreover, we can add to the shared leaf an additional level of tests to account for additional components of effects-sharing, such as a brain tissue-shared component.

### Comparison to other methods

We compared the prediction accuracy of CONTENT to a context-by-context TWAS model[3,4] in which the expression of each context is modeled separately, and to UTMOST[9], a method that jointly learns the genetic effects on all contexts simultaneously. Specifically the model based on TWAS fits a penalized linear model for each context. UTMOST, on the other hand, employs a group LASSO penalty across all contexts simultaneously, allowing it to gain power over the context-by-context approach by considering all individuals and contexts in a study at once. As we were we able to use a fast R package for penalized regression[80], we used 10-fold cross-validation to fit the context-by-context model. Owing to UTMOST's computational intensity, we used its default value of 5 folds for cross-validation.

We also compared CONTENT to a previous approach by Wheeler et al., orthogonal tissue decomposition (OTD)[14]. OTD is a direct correlate of CONTENT(Shared) and CONTENT(Specific), and is generated by fitting a mixed effects model across all contexts for a given individual. Namely, a mixed effects model is fit as follows: an individual's expression across all tissues is set as the outcome, the shared expression is modeled as a random individual-level intercept and is estimated using the posterior mean, and the specific expression is treated as the residuals from

the fit model (after adjusting for covariates). Under infinite sample sizes, the components of OTD are equivalent to CONTENT(Shared) and CONTENT(Specific).

### Evaluations on GTEx and CLUES

We residualized the expression of each gene in each context over their corresponding covariates (e.g., PEER factors, age, sex, batch information) prior to fitting UTMOST and an elastic-net model for each context in the context-by-context approach. We did the same residualization before decomposing and then fitting the context-shared and context-specific components with an elastic net for CONTENT. After generating cross-validated predictors for each method, we examined the number of significantly predicted genes as well as the prediction accuracy (in terms of adjusted $R^2$) between the cross-validation-predicted and true gene expression per gene-context pair.

To properly control the false discovery proportion at .05 across-contexts and within-methods, we employed a hierarchical FDR correction[17,18] separately for CONTENT, UTMOST, and the context-by-context approaches. Notably, using this correction for all methods provides a generous comparison to previous methods, as when there exists at least one significantly heritable gene-context association for a given gene, there is a relative gain in power over the context-by-context FDR for other contexts tested within this gene[17,18].

### Application to TWAS

We performed transcription-wide association studies across 24 phenotypes using FUSION-TWAS[3]. FUSION-TWAS uses GWAS summary statistics and user-specified gene expression weights with an LD reference panel to perform the test of association between genetically predicted gene expression and a phenotype of interest. We tested a gene-context pair for association if the pair's expression was predicted at a nominal $p$-value of .1, and note that this threshold does not substantially alter the number of TWAS discoveries (Supplementary Fig. 16). Notably, previous methods may use their own test of gene-context-trait association or leverage set tests (e.g. Berk Jones[9]) to combine their associations across all contexts for a given gene and therefore increase power. In this comparison, we report the association as output by FUSION (a single gene-context-trait association) and corrected by hierarchical false discovery without any sort of set test for the sake of equality in the comparison. We ran FUSION-TWAS using the default recommended settings, with reference data from the 1000 genomes project[81]. TWAS weights were trained on the GTEx v7 dataset[2] as well as the CLUES[20] single-cell RNAseq dataset of PBMCs. For a given gene-context-trio, we ran up to 5 TWAS—1) context-by-context, 2) UTMOST, 3) CONTENT(Shared), 4) CONTENT(Specific), and 5) CONTENT(Full). Notably, we re-trained each methods' predictors on genetic variants that are present in the LDREF cohort as well as GTEx or CLUES to ensure selected expression weights had overlap with the reference panel (LDREF).

### Simulations to evaluate prediction accuracy

To evaluate the properties of our method relative to other methods we perform a series of simulation experiments. We first simulate genotypes for each individual, where each individual $i$ and each locus $m$ ($m = 1:M$) is independent, and there are no rare SNPs:

$$\mathbf{G}_{im} \sim \text{Bin}(2, \text{Unif}[0.05, 0.50]) \tag{11}$$

We then draw both context-shared ($\beta_{j.}$) and context-specific ($\beta_{jc}$) effect sizes for each SNP from a normal distribution with a Bernoulli random variable $I_m$ controlling the probability that the $m^{th}$ SNP is

causal (i.e. induce sparsity of genetic effects).:

$$I^m \sim \text{Bernoulli}(.05), \beta_{j.}^m \sim \text{N}\left(0, \frac{h^2}{M*\pi}\right) \times I^m, \text{ and } \beta_{jc}^m \sim \text{N}\left(0, \frac{h_c^2}{\lambda*M*\pi}\right) \times I_\lambda^m \quad (12)$$

Here, $h^2$ and $h_c^2$ are the context-shared and context-specific heritabilities of expression on gene $j$. In general, the SNPs with nonzero context-specific effect sizes were subsampled from SNPs with nonzero context-shared effect sizes. We additionally simulate for a subset of contexts some number of truly context-specific eQTLs drawn from Poisson($\lambda = 1$) for randomly selected SNPs that were not eQTLs for the context-shared effects. Finally, we simulate the expression of gene $j$ as follows:

$$\mathbf{E}_{jc} = \mathbf{G}_j \beta_{j.} + \mathbf{G}_j \beta_{jc} + \varepsilon_{jc} \quad (13)$$

$$\varepsilon \sim \mathcal{N}(\mathbf{0}, \boldsymbol{\Sigma}), \ \boldsymbol{\Sigma} \in \mathbb{R}^{C \times C} = \begin{bmatrix} \sigma_1^2 & \dots & \sigma_{1,C} \\ \vdots & \ddots & \vdots \\ \sigma_{C,1} & \dots & \sigma_C^2 \end{bmatrix} \quad (14)$$

where $\varepsilon \in \mathbb{R}^I$, represents the correlation of environment or intra-individual noise across contexts, $\sigma_c^2 = 1 - h^2 - h_c^2$ is the variances of each context $c$, and $\sigma_{c_1,c_2} = \rho_{c_1,c_2} \sigma_{c_1} \sigma_{c_2}$ is the covariance of context $c_1$ and $c_2$. We generated data under varying levels of context-specific heritability, truly context-specific eQTLs, causal SNPs, and correlation of intra-individual noise across contexts. The number of contexts was set to 20, and to replicate a setting similar to GTEx, the corresponding sample sizes of each ranged from 75 to 410 where individuals were not necessarily measured in every context. In our simulations, we generated one train and one test data set using the above framework. We evaluated the performance of each method by comparing the true and predicted expression in the test data set, using the predictor learned from the training data set.

To assess the effect of additional sharing on a subset of contexts, we also set up a simulation framework using the same generative process as above, only that a subset of contexts also received additional genetic effects. More rigorously, for this subset of contexts (acting as brain contexts in GTEx, for example), expression was generated as in equation (6) with an additional term:

$$\mathbf{E}_{jc} = \mathbf{G}_j \beta_{j.} + \mathbf{G}_j \beta_{jc} + \mathbf{G}_j \beta_{jb} + \varepsilon_{jc}, \quad \beta_{jb}^m \sim \text{N}\left(0, \frac{h_b^2}{\lambda*M*\pi}\right) \times I_\lambda^m \quad (15)$$

where each variable is simulated as before, $\beta_{jb}^m$ corresponds to additional genetic effects that are subsampled from SNPs that have a context-shared effect, and $h_b^2$ is the brain-shared heritability.

### Simulations of TWAS performance

Using the above generated genotypes and gene expression, we simulated phenotypes to evaluate the performance of each method under the assumed model in TWAS. For a given phenotype, we randomly selected 300 gene-context pairs (100 genes, 3 contexts each) whose expression would comprise a portion of a phenotype. Explicitly, we generated a phenotype as follows:

$$y_i = E_i \delta + \varepsilon \quad \delta \sim N\left(0, \frac{\sigma_{ge}^2}{300}\right), \ \varepsilon_i \sim N\left(0, 1 - \frac{\sigma_{ge}^2}{300}\right) \quad (16)$$

Where $E_i$ is the standardized genetic expression of the 300 gene-context pairs for individual $i$, $\delta$ is the length-300 vector of effect sizes for each gene-contexts' expression, $\sigma_{ge}^2$ is the variance in the phenotype $y_i$ due to cis-genetic gene expression, and $\varepsilon_i$ corresponds to environmental effects (or noise) as well as trans-

genetic effects for individual $i$. In our simulations, we varied the heritability of gene expression and fixed variability in the phenotype due to genetic gene expression to 0.2. To simulate a wide range of genetic architectures, the proportion of heritability of gene expression due to the context-shared effects was sampled from a standard uniform distribution, and the proportion of heritability due to context-specific effects was (1- the context-shared proportion). Once we generated a phenotype, we performed a TWAS using weights output from each method by imputing expression into a simulated external, independent set of 10000 genotypes that followed the same generation process as in the previous subsection.

### Reporting summary
Further information on research design is available in the Nature Research Reporting Summary linked to this article.

### Data availability
GTEx v7 is a publicly available dataset through the GTEx portal (genotypes must be accessed through dbGap permissions, and RNA sequencing is available on the GTEx website; https://www.gtexportal.org/home/datasets[11]). The CLUES dataset is also publicly available[21] at Gene Expression Omnibus accession number GSE174188 and dbGap accession number phs002812.v1.p1. Trained weights for the GTEx v7 dataset and our in-house single-cell RNAseq are available at the TWAS/FUSION repository (http://gusevlab.org/projects/fusion/). We provide TWAS summary statistics for all three methods on both datasets (as well as an indicator of whether the association was hierarchical FDR-adjusted significant) at Zenodo accession https://doi.org/10.5281/zenodo.5209239. We include summary statistics for the associations within GTEx and CLUES at the above Zenodo link.

### Code availability
The CONTENT software is freely available at https://github.com/cozygene/CONTENT. We include in the same link an example script for running hFDR.

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

## Acknowledgements

M.T. is supported in part by NIH Training Grant in Genomic Analysis and Interpretation T32HG002536. N.Z. was funded by NIH, CZI, and V.A. grants U01HG012079, U01MH126798, R01MH125252, 1R01HG011345, U01HG009080, CZF2019-002449, R01ES029929, R01HL155024, 1I01CX002011. B.B. received support from U01HG012079. This work was also funded by the National Science Foundation (Grant No. 1705197), and by NIH/NHGRI HG010505-02. C.J.Y. received funding from NIH grants P30AR070155, R01AR071522, U01HG012192, R21AI133337, and CZI P0535277. M.G.G. was supported by NIH grant 1F31HG011007. Human and organ icons in Fig. 1 were created by Biorender.com and further edited individually; the entire illustration was made using Microsoft Powerpoint.

## Author contributions

N.Z. and B.B. conceived of the project and developed the statistical methods with M.T.; M.T. implemented the comparisons with simulated data with contributions from A.T.; M.T., A.L., and M.G.G., performed the analyses of the GTEx and CLUES data and additional analyses. M.T. implemented the software. M.T., N.Z., and B.B. wrote the manuscript, with significant input from E.H., C.J.Y., A.G., and M.G.G.; A.G. prepared the online data resources.

## Competing interests

C.J.Y. is a Scientific Advisory Board member for and holds equity in Related Sciences and ImmunAI. C.J.Y. is a consultant for and holds equity in Maze Therapeutics. C.J.Y. is a consultant for TReX Bio. C.J.Y. has received research support from Chan Zuckerberg Initiative, Chan Zuckerberg Biohub, and Genentech. E.H. is senior vice president of AI/ML at OptumLabs (Minnetonka, MN). The remaining authors report no competing interests.

## Additional information

[1]Department of Computer Science, University of California Los Angeles, Los Angeles, CA, USA. [2]Department of Bioengineering and Therapeutic Sciences, University of California, San Francisco, San Francisco, CA, USA. [3]Institute for Human Genetics, University of California, San Francisco, San Francisco, CA, USA. [4]Biological and Medical Informatics Graduate Program, University of California, San Francisco, San Francisco, CA, USA. [5]UCLA-Caltech Medical Scientist Training Program, David Geffen School of Medicine, University of California Los Angeles, Los Angeles, CA, USA. [6]Department of Mathematics, Indian Institute of Technology Delhi, Hauz Khas, Delhi, India. [7]Department of Human Genetics, University of California Los Angeles, Los Angeles, CA, USA. [8]Department of Anesthesiology and Perioperative Medicine, University of California Los Angeles, Los Angeles, CA, USA. [9]Department of Computational Medicine, University of California Los Angeles, Los Angeles, CA, USA. [10]Department of Medical Oncology, Dana-Farber Cancer Institute and Harvard Medical School, Boston, MA, US. [11]Division of Genetics, Brigham and Women's Hospital, Boston, MA, US. [12]Chan-Zuckerberg Biohub, San Francisco, CA, USA. [13]Division of Rheumatology, Department of Medicine, University of California, San Francisco, San Francisco, CA, USA. [14]Institute for Computational Health Sciences, University of California, San Francisco, San Francisco, CA, USA. [15]Department of Neurology, University of California Los Angeles, Los Angeles, CA, USA.
✉e-mail: mjthompson@ucla.edu; nzaitlen@ucla.edu

