## [Peer Review File · Nature Communications]

Multi-context genetic modeling of transcriptional regulation resolves novel disease lociREVIEWER COMMENTS

Reviewer #1 (Remarks to the Author):

Thompson et al propose to build on transcriptome-wide association studies (TWAS), addressing some of the limitations of existing implementations. The authors generate context-specific and context-shared models of the genetic component of gene expression, leveraging the correlation structure present in multi-context studies. They report that their TWAS method -- CONTENT -- leads to a substantial increase in the number of genetically predicted genes (in both bulk-tissue and single-cell scenarios) as well as in the number of gene-phenotype associations discovered relative to existing methods. Using CONTENT, they show improved prediction accuracy of the genetic contribution to expression variability while exploiting some of the key findings of recent GTEx studies on the tissue specificity/sharedness of the regulatory architecture of gene expression. Methodologically, the aims of the study are significant and can have an impact on the field if the claims are substantiated.

The study contains some notable findings and insights. However, the following major concerns should be addressed.

1. The models for the tissue-specific and tissue-shared components must be made publicly available. I followed the link to the twas-hub (from the Data availability section), but the models were not accessible, which made it impossible to evaluate some of the authors' claims. (The software repository (Github) for the project did not have the models either.)
2. Although the authors provide performance comparisons across methods, I think it will be important to show a direct comparison -- across the genes -- of the performance (e.g., cross-validation R²) from the study's shared, specific and full models with the existing methods, i.e., the tissue-by-tissue and UTMOST approaches. (It would be useful to present these comparisons in scatter plots. What genes are showing substantial differential performance?)
3. There has to be substantial clarification of how the models are applied to GWAS summary statistics to identify gene-trait associations. The issue of LD and the presence of the 3 models must be clarified in application to GWAS data.
4. It is not clear how horizontal pleiotropy or LD contamination (well-known major issues with TWAS interpretation) is methodologically addressed by the proposed method.
5. There needs to be better clarification or discussion of what the study adds to the literature. As the authors have indicated throughout, there is already an orthogonal decomposition method (Wheeler et al.). The hierarchical FDR approach is also already published. This should be part of the Discussion.

Reviewer #2 (Remarks to the Author):

The authors describe an approach to identifying causal genes based on decomposing gene expression into tissue shared and tissue specific, and creating a new expression phenotype which combines tissue shared with all of the tissue specific components. They demonstrate that using this approach identifies significantly more genes than methods that look at tissues separately or methods that combine tissues to leverage shared signals. The paper is well written and the authors clearly explain their advances over current practice. My major concern is that the major justification for their method is in the greater numbers of genes implicated, they have not demonstrated that these new genes are not false

positives relating to statistical artefacts.

It is well known that issues such as pleiotropy and linkage contamination mean that TWAS can often suggest genes that have no causal relationship with the disease in question (Wainberg et al, 2019, Nature Genetics). The authors have not shown that the novel genes they discover are not concentrated in these false positives. Ndungu et al, 2020, AJHG dug deeper into this issue. By building "full" models which include SNPs related to all of the underlying expression phenotypes my concern would be that this set of SNPs is more likely to be in LD with a very strong GWAS hit and this overlap drives significance, creating false positives. To allay this concern the authors could: apply their method to a trait where the "ground truth" is known, such as the metabolites from Ndungu et al, b) demonstrate that statistical significance is not driven by the number of SNPs in the gene model, and/or c) produce independent evidence that the novel genes they propose are trait related.

The authors considerably overstate the issues with controlling the false discovery rate separately for each context or using the Bonferroni approach across all tests (lines 38-40). The false discovery rate is a proportion, if the proportion of false positives within each context is controlled then the proportion of false positives over all tests should also be controlled. To quote the Efron 2008 paper that discusses the issue: "False discovery rate methods are more forgiving—usually they can be applied to separate analyses in unchanged form without undermining their inferential value." As for the Bonferroni correction over all tests being too stringent: it is as stringent as it needs to be if the probability of declaring even one false positive is to be controlled. A hierarchical FDR adjustment will not achieve that. The statement: "existing methods employ multiple testing strategies that either fail to control for all tests performed, (e.g., by controlling the false discovery rate (FDR) within each context separately [4, 15]), or act too stringently (e.g., by using Bonferroni adjustment across all contexts [15])" I see as factually incorrect.

I'm sure there are other justifications that can be raised for applying the hierarchical FDR approach. My concern would be that in a paper which introduces both a new way of building gene models and a new way of controlling for multiple testing can confuse the reader on from where exactly any advances originate. It is not clear to me that hFDR is applied to the context specific analysis comparator.

The authors state that genetic effects are shared to a greater extent across tissues (liver to brain for example) than across blood cell types. To me this is highly counterintuitive. Is it possible to investigate this more thoroughly, could it be an artefact of the technologies used to quantify bulk vs single cell?

The authors state that their method is orders of magnitude faster than others, I think it would be good to include an example to place this in context.

We thank the reviewers for their time and feedback. We appreciate the commentary and suggestions raised by the reviewers and believe we have addressed the concerns raised. Namely, we have included multiple additional TWAS analyses exploring the issues of LD and horizontal pleiotropy when running CONTENT. We also show a novel comparison of the differences in discoveries made when using a traditional FDR in comparison to our hierarchical FDR (hFDR) implementation. Moreover, we have made substantial changes to the language in the text to more clearly explain our analyses, methods used, and novel contributions to the field. We believe the manuscript is now more cohesive and presents a more thorough report of our findings. We first present the full list of new experiments conducted and then provide detailed responses in line and in blue below.

The modifications and additions include:

1. Replication and evaluation of an analysis by Ndungu et al. (AJHG 2020) for fine-mapping.
2. Showing the improved ability of CONTENT over previous approaches to discover the correct, known causal gene in TWAS.
3. Displaying the performance of CONTENT and previous approaches on a subset of tissues and tissue-gene pairs to examine under which genetic architectures each method is likely to perform best.
4. Comparing the number of discoveries made with and without hierarchical FDR adjustment.
5. Demonstrating that our hFDR implementation can boost discoveries made for tissues with relatively smaller sample sizes.
6. Showing that the number of TWAS gene discoveries by the CONTENT(Full) model is independent of the number of non-zero weights in its model.
7. Adding further discussion on issues of LD and horizontal pleiotropy.
8. Clearly describing our novel contributions to the field, including the first scRNA-seq TWAS models, as well as how our findings and method will be useful.

Reviewer #1 (Remarks to the Author):

The study contains some notable findings and insights. However, the following major concerns should be addressed.

1. The models for the tissue-specific and tissue-shared components must be made publicly available. I followed the link to the twas-hub (from the Data availability section), but the models were not accessible, which made it impossible to evaluate some of the authors' claims. (The software repository (Github) for the project did not have the models either.)

We thank the reviewer for bringing this to our attention. Our intention was to have the weights available while they were being processed through the TWAShub analysis pipeline, and we had overlooked this when submitting the manuscript. The weights are now available through the TWAS/FUSION software repository at <http://gusevlab.org/projects/fusion/>. As the TWAShub

pipeline is computationally demanding, we plan to have our results available on the TWAShub resource when the paper is finalized and no further modifications to the weights or analysis need to be made. We have changed the language in the text to point readers to the FUSION link.

2. Although the authors provide performance comparisons across methods, I think it will be important to show a direct comparison -- across the genes -- of the performance (e.g., cross-validation R²) from the study's shared, specific, and full models with the existing methods, i.e., the tissue-by-tissue and UTMOST approaches. (It would be useful to present these comparisons in scatter plots. What genes are showing substantial differential performance?)

We agree with the reviewer there may be interesting differences in performance across genes. In our simulations, we found that each method (CONTENT, context-by-context and UTMOST) may perform more favorably than others depending on the underlying genetic architecture. Empirically, we found that UTMOST performs best when the underlying causal genetics is shared across contexts, and that the context-by-context approach performs best when the underlying causal genetics is more specific to each context. CONTENT outperformed both models as it explicitly accounts for both types of heritability. Moreover, in GTEx we found that UTMOST on average explained more variability than the context-by-context approach for contexts with relatively small sample sizes, but that it did not perform as well in contexts with relatively larger sample sizes. On average, CONTENT performed much better than both methods. We annotated several genes for which the performance was quite different between CONTENT and either the context-by-context or UTMOST approaches. Generally, these genes were in line with our expectations set by our simulation results as described above (shown below). Nonetheless, we agree this was an illustrative figure and now include it in the supplementary materials (Figure S11).

In terms of visualizations, we feel a broad level of comparisons (20,000+ genes X 48 contexts X 3 methods) may be difficult to succinctly portray in the manuscript. As such, we have included a small selection of scatterplots comparing the performance of the context-by-context approach, UTMOST and the best cross-validated CONTENT model on a representative selection of tissues (from relatively small, moderate and large sample sizes). We also now include as a supplementary file a spreadsheet with the performance of each method (including all three CONTENT models) across all genes and contexts.

Additional discussion of individual genes is provided in the figure caption of Supplementary Figure 11.

3. There has to be substantial clarification of how the models are applied to GWAS summary statistics to identify gene-trait associations. The issue of LD and the presence of the 3 models must be clarified in application to GWAS data.

We have added to the Methods section a more complete description of the steps we took to run TWAS (Page 24). Briefly, as input, TWAS-FUSION takes a set of GWAS summary statistics, an LD reference panel, and a set of weights used to predict gene expression for each model. For each gene we ran up to 5 TWAS models: 1) context-by-context, 2) UTMOST, 3) CONTENT(Shared), 4) CONTENT(Specific), and 5) CONTENT(Full). We only ran models whose performance in the training data exceeded a threshold. TWAS-FUSION then outputs summary statistics (e.g., p-value, z-score) for each model run. We then apply a multiple test correction adjustment FDR or hFDR to account for the multiple models and tissues examined.

We also include a more thorough response to the issue of LD in the paper and in our response to point 4 below.

4. It is not clear how horizontal pleiotropy or LD contamination (well-known major issues with TWAS interpretation) is methodologically addressed by the proposed method.

We appreciate the reviewer raising this concern. We realize the language in our original text was not clear enough. We do not claim that our method addresses the issue of horizontal pleiotropy or LD contamination better than UTMOST or context-by-context approaches. Instead, like TWAS, Predixcan, UTMOST, etc, the goal is to identify loci containing causal associations. We believe we have accomplished this goal better than previous methods, aside from concerns about horizontal pleiotropy or LD contamination—there are now more loci available at which to discover genuine, causal genes in downstream analyses. (CONTENT discovered 1834 loci with GTEx weights and 735 with CLUES weights compared to 1247 and 463 with the context-by-context approach and 823 and 382 with UTMOST.) See page 17 for clarification and discussion in the manuscript.

The issues raised by horizontal pleiotropy and LD contamination are not unfounded, but this is a natural effect of any method that builds more powerful TWAS predictors. Our increase in variance explained is analogous to gains that would be made with using only the context-by-context model, and increasing the sample size substantially. In other words, if we limited the discussion to the context-by-context approach alone, such an increase in variance explained (and therefore more models built) would be a result of increasing the sample size and thus power. One advantage of the CONTENT model can therefore be thought of as gaining power in a fashion akin to increasing sample size. We do not believe that using a less powerful method (or purposely removing samples) is a strong argument for combating LD contamination. Just like in the context of GWAS fine-mapping, resolution should increase with better models and larger sample sizes. Finally, having more models because the method is more powerful should also increase the chance that the causal gene is in fact included in the analyses.

Understandably, reviewer #2 raised a similar concern. They requested we revisit an analysis of metabolite summary statistics for which the causal gene may be known. Though CONTENT introduces more models to be tested, it added to the number of known causal genes discovered. Furthermore, we saw a slight improvement in the rank of the causal gene (which is often the top gene) relative to previous approaches, demonstrating that CONTENT is on par or

slightly better than previous approaches in terms of LD contamination. Please see our response below for more details as well as the text on pages 14-15.

There exist several approaches for dealing with LD contamination and horizontal pleiotropy on TWAS summary statistics[1,7,8], and are fully compatible with CONTENT output.

Lastly, we wish to emphasize that groups commonly construct methods for gaining power in GWAS or for improving power in variant fine-mapping. It is uncommon that a methods paper would propose a solution for both problems of association and fine-mapping, and we argue that it would be atypical of a TWAS paper to build both better predictors and suggest a fine-mapping strategy. Indeed this is generally the case[1-6], with rare exceptions[7].

We agree that this is an important point of discussion and have therefore added related commentary to the Discussion section of the manuscript.

[1] Gusev A., et al. Integrative approaches for large-scale transcriptome-wide association studies. *Nat Genet.* 2016 Mar;48(3):245-52. doi: 10.1038/ng.3506.

[2] Gamazon E., et al. A gene-based association method for mapping traits using reference transcriptome data. *Nat Genet* 47, 1091–1098 (2015). <https://doi.org/10.1038/ng.3367>

[3] Barbeira A. et al. (2019) Integrating predicted transcriptome from multiple tissues improves association detection. *PLOS Genetics* 15(1): e1007889. <https://doi.org/10.1371/journal.pgen.1007889>

[4] Hu Y., et al. A statistical framework for cross-tissue transcriptome-wide association analysis. *Nat Genet.* 2019 Mar;51(3):568-576. doi: 10.1038/s41588-019-0345-7.

[5] Nagpal S., TIGAR: An Improved Bayesian Tool for Transcriptomic Data Imputation Enhances Gene Mapping of Complex Traits. *Am J Hum Genet.* 2019 Aug 1;105(2):258-266. doi: 10.1016/j.ajhg.2019.05.018.

[6] Parrish R., et al. TIGAR-V2: Efficient TWAS tool with nonparametric Bayesian eQTL weights of 49 tissue types from GTEx V8. *HGG Adv.* 2021 Nov 4;3(1):100068. doi: 10.1016/j.xhgg.2021.100068.

[7] Zhou D., et al. A unified framework for joint-tissue transcriptome-wide association and Mendelian randomization analysis. *Nat Genet.* 2020 Nov;52(11):1239-1246. doi: 10.1038/s41588-020-0706-2.

[8] Mancuso N., et al. Probabilistic fine-mapping of transcriptome-wide association studies. *Nat Genet.* 2019;51(4):675-682. doi:10.1038/s41588-019-0367-1

5. There needs to be better clarification or discussion of what the study adds to the literature. As the authors have indicated throughout, there is already an orthogonal decomposition method (Wheeler et al.). The hierarchical FDR approach is also already published. This should be part of the Discussion.

In light of this comment, we have elaborated further on this in the Discussion section (Page 18) and include our thoughts as follows. The decomposition method proposed by Wheeler is based on a linear mixed model. Our approach is much simpler, in that it only requires calculating a sample mean and subtracting the observed values to generate its predictors. Additionally, and more importantly, Wheeler et al did not offer a way to combine their predictors, which we show is critical for improving power in downstream TWAS. Moreover, our approach extends the Wheeler et al method by making it adaptable to additional levels of pleiotropy as we show in the supplementary methods (Page S26) .

Although hFDR is already published, it is not currently being used in the TWAS literature, and the structure of the hierarchy must be adapted to this application, especially when multiple methods are considered. We believe our model (and inclusion of other models for TWAS) provides a natural example of when such an approach may be useful, and are hopeful that other investigations will make use of the same framework. Moreover, in practice, people often fit models using multiple methods for performing TWAS. We have laid the framework in this text for how to properly control the FDR when doing so. Nonetheless, we agree that this specific contribution may be minor and have reduced our claims about it throughout the text.

Outside of our methodological and practical contributions, we also add to the literature an in-depth overview of where different TWAS methods flourish (i.e. under which genetic architectures one may expect to gain power by using a certain TWAS method) as well as a novel evaluation of the difference of bulk and single cell RNA seq. We believe **this is the first single cell RNA-seq TWAS analysis to date**. We show through multiple experiments, scRNA-seq TWAS provide different conclusions about the underlying genetic architecture. In our eGene and TWAS analyses, we show that single cell expression data may harbor more specific genetic variability than bulk expression data (which is dominated by shared genetic variability). To our knowledge, this was not previously appreciated, and we believe that it will be a very interesting finding for the field. We emphasize that our decomposition and CONTENT(Full) model were critical for this finding.

Reviewer #2 (Remarks to the Author):

It is well known that issues such as pleiotropy and linkage contamination mean that TWAS can often suggest genes that have no causal relationship with the disease in question (Wainberg et al, 2019, Nature Genetics). The authors have not shown that the novel genes they discover are not concentrated in these false positives. Ndungu et al, 2020, AJHG dug deeper into this issue. By building "full" models which include SNPs related to all of the underlying expression phenotypes my concern would be that this set of SNPs is more likely to be in LD with a very strong GWAS hit and this overlap drives significance, creating false positives. To allay this concern the authors could: apply their method to a trait where the "ground truth" is known, such

as the metabolites from Ndungu et al, b) demonstrate that statistical significance is not driven by the number of SNPs in the gene model, and/or c) produce independent evidence that the novel genes they propose are trait related.

We appreciate the reviewer's concern and note that Reviewer #1 also raised a similar point. We emphasize that not only do we discover more genes than previous approaches, but also more independent loci for each disease (CONTENT discovered 1834 loci with GTEx weights and 735 with CLUES weights compared to 1247 and 463 with the context-by-context approach and 823 and 382 with UTMOST), which responds to point (c) above as these loci must each have at least one causal trait association. We point the reviewer to our response to Reviewer #1 for further discussion and detail.

We next address point (b). We emphasize that the full model of CONTENT is simply the element-wise sum of weights from the specific and shared models of CONTENT. We examined for each CONTENT(Full) TWAS the correlation between the absolute value of TWAS Z scores and the number of non-zero weights in the corresponding model and found that the correlation is negligible across phenotypes (average -0.01965 , sd $.00998$). We show an example below and conclude that the number of genes discovered in a TWAS using CONTENT is independent of the number of SNPs with non-zero weights in our predictive models.

In response to point (a), we conduct and examine a replication of the Ndungu et al. paper. These authors cite a GWAS paper by Shin et al., in which they discover a collection of independent SNPs that are associated with a collection of metabolites. Shin et al. evaluate evidence for genes near their variant loci and determine a set of predicted causal genes based on the known gene-protein product and metabolite relationships. We pulled data from Shin et al. and we were able to find a collection of 58 gene-trait predicted-causal pairs on which to evaluate CONTENT and the context-by-context approach. While we do not believe this is necessarily a causal set of genes, we proceed with the analysis as requested.

After running TWAS and hFDR for both CONTENT and the context-by-context approach, CONTENT discovered more of the predicted causal genes (39 compared to 36 for the context-by-context approach of the 58 known gene-metabolite pairs; Supplementary Table S1). We note that this is in line with our applications of TWAS to GTEx and CLUES, despite CONTENT paying a larger multiple testing penalty since its total number of models is greater than that of the context-by-context approach.

Moreover, we conducted an analysis to evaluate how each method ranked the causal gene relative to other associated genes within the same locus. To reiterate, Ndungu et al. state that many genes can act as “passenger” genes in the sense that within a locus, non-causal genes are unintentionally called statistically significant owing to LD contamination or horizontal pleiotropy. We wished to see if the extent to which CONTENT suffers from this phenomenon was relatively greater than the context-by-context approach as CONTENT potentially implicates more genes at a given locus (due to the fact that it is a more powerful predictor, and therefore builds more models). In our analysis, we used the maximum absolute Z score for a gene, and ranked all genes within the locus (+- 1Mb) of the discovered causal gene to see where the causal gene ranks (i.e. if it has the highest Z score within its locus). Despite having more models built per locus, CONTENT ranked the known causal gene slightly better in comparison to the context-by-context approach on the intersection of gene-metabolite pairs discovered by both methods (CONTENT average rank of 2.257 compared to context-by-context rank of 2.371, where a ranking of 1 is ideal).

We thank the reviewer for pointing us to this manuscript and analysis. We have now included this analysis in our Results section (pages 14-15) as well as some commentary on this topic in the Discussion section (page 17).

The authors considerably overstate the issues with controlling the false discovery rate separately for each context or using the Bonferroni approach across all tests (lines 38-40). The false discovery rate is a proportion, if the proportion of false positives within each context is controlled then the proportion of false positives over all tests should also be controlled. To quote the Efron 2008 paper that discusses the issue: "False discovery rate methods are more forgiving—usually they can be applied to separate analyses in unchanged form without undermining their inferential value." As for the Bonferroni correction over all tests being too stringent: it is as stringent as it needs to be if the probability of declaring even one false positive is to be controlled. A hierarchical FDR adjustment will not achieve that. The statement: "existing methods employ multiple testing strategies that either fail to control for all tests performed, (e.g., by controlling the false discovery rate (FDR) within each context separately [4, 15]), or act too stringently (e.g., by using Bonferroni adjustment across all contexts [15])" I see as factually incorrect.

We agree that we need to modify our language in the text. We have done so as stated below. Nonetheless, while we agree it is counter-intuitive that the FDR is not the same despite the number of contexts, the inventors of the FDR (and hFDR) state that this is not the case when there is a relationship amongst the collection of hypothesis tests. For example, the relationship of testing the prediction of the same gene across contexts. Briefly, the *average* FDR across contexts will be controlled, but the total, global FDR as well as the FDR within contexts will not be[1].

We also investigated the extent to which controlling the FDR within each tissue and method separately changes the number of significant associations relative to controlling the FDR simultaneously across contexts and methods with hFDR. The changes are most interpretable using the context-by-context approach, as there is no “borrowing” of power by tissues with a small sample size from tissues with a larger sample size (which may be the case with CONTENT and UTMOST). Interestingly, we see changes that are consistent with the hFDR manuscript: in tissues with small sample sizes, there are fewer significant associations when using FDR rather than hFDR. Conversely, in tissues with larger sample sizes, there are relatively greater numbers of significant associations. We now present these results in the supplementary text (Table S1 and Figure S12).

“existing methods with the goal of maximizing the number of discoveries made may employ multiple testing strategies that either fail to control the false discovery rate properly (e.g., by using FDR within each context separately [4, 15]), or limit their discoveries as they are based on conservative FWER control (e.g. Bonferroni adjustment across all contexts [15]).”

[1] Peterson, C. B., et al. (2016). Many Phenotypes Without Many False Discoveries: Error Controlling Strategies for Multitrait Association Studies. *Genetic epidemiology*, 40(1), 45–56. <https://doi.org/10.1002/gepi.21942>

I'm sure there are other justifications that can be raised for applying the hierarchical FDR approach. My concern would be that in a paper which introduces both a new way of building gene models and a new way of controlling for multiple testing can confuse the reader on from where exactly any advances originate. It is not clear to me that hFDR is applied to the context specific analysis comparator.

We appreciate the reviewer mentioning this. We have done our best to edit the language throughout the text to distinguish our advances. Reviewer #1 brought up a similar concern, and we have added more to the Discussion section to highlight the novelty of our work. We also point the reviewer to the response above.

Indeed, we have employed hFDR for all methods (CONTENT, context-by-context, and UTMOST). Notably, though, CONTENT has an extra level on the hierarchy, gene-tissue-method as opposed to gene-tissue, and we have done our best to clarify this in the text. Finally, we have a table comparing FDR and hFDR results (Table S1) that allows the reader to see how much of the improvement comes from hFDR vs the CONTENT model.

The authors state that genetic effects are shared to a greater extent across tissues (liver to brain for example) than across blood cell types. To me this is highly counterintuitive. Is it possible to investigate this more thoroughly, could it be an artefact of the technologies used to quantify bulk vs single cell?

We also think that this is an interesting finding and believe the source of this issue is driven by the high degree of sharing of cell types across contexts, which has been hypothesized before in analyses of GTEx[1]. For example, Endothelial cells can be found in Breast, Endometrium, Esophagus, Eye, Heart muscle, Liver, Lung, Ovary, Pancreas, Placenta, Prostate, Skeletal muscle, and Skin and often make up a substantial fraction of the collected tissue:

<https://www.proteinatlas.org/humanproteome/single+cell+type>[2,3].

We believe our work is consistent with this observation. In brief, eQTLs can be cell-type specific (meaning that they only appear in a few or a single cell type) and may appear as shared across tissues composed of a high proportion of the cell-type(s) in which the eQTL is present. We also found that the proportion of eGenes that also had a shared component of expression was substantially lower at the single-cell level relative to the bulk, tissue level. What's more is that the ability to discover context-specific components of expression is indeed related to sample size in the GTEx dataset. Despite the above, and having a lower number of individuals in the single-cell data, we discover a greater proportion of genes with a context-specific component than in GTEx. Further, when there exists a CONTENT(Full) model, it is dominated by the specific variability at the single-cell level, whereas it is dominated by the shared variability at the tissue level.

These findings may be due to the fact that the context-specific signal is much stronger at the single-cell level, and is therefore easier to capture in a smaller sample size. This may also be consistent with complex patterns of celltype heterogeneity—if celltypes harbor more specific

variability than shared variability, then comparing the level of sharing between, say, cardiac and liver cells should be low. However, since bulk sequencing very likely includes many white blood cells (in addition to the target tissue celltypes), the estimate of this sharing is also likely to be inflated. We have added a bit more on these points to the Discussion section (Page 18).

[1] Kim-Hellmuth, Sarah et al. "Cell type-specific genetic regulation of gene expression across human tissues." *Science (New York, N.Y.)* vol. 369,6509 (2020): eaaz8528.

doi:10.1126/science.aaz8528

[2] Karlson M., et al. A single-cell type transcriptomics map of human tissues. *Sci Adv.* 2021 Jul 28;7(31):eabh2169. doi: 10.1126/sciadv.abh2169. PMID: 34321199; PMCID: PMC8318366.

[3] Uhlen M., et al. Proteomics. Tissue-based map of the human proteome. *Science.* 2015 Jan 23;347(6220):1260419. doi: 10.1126/science.1260419. PMID: 25613900.

The authors state that their method is orders of magnitude faster than others, I think it would be good to include an example to place this in context.

We agree. In addition to showing the computational runtime of our software compared to older methods (Supplementary Figure S9), we also include a citation to FastGxE[1], the framework on which CONTENT was built. FastGxE shows that the decomposition framework is substantially faster than fitting traditional LMM approaches:

"When extrapolated to mapping cis-eQTLs in the entire GTEx dataset, i.e. approximately 200M tests for 25K genes and 3M SNPs, we found that LMM-GxC and LM-GxC would finish in approximately 30 years and 10 months, respectively, while CxC and FastGxC achieved equivalent results in under one minute (average run time in 100 iterations)."

[1] Lu A., et al. Fast and powerful statistical method for context-specific QTL mapping in multi-context genomic studies. *bioRxiv* 2021.06.17.448889; doi:

<https://doi.org/10.1101/2021.06.17.448889>

REVIEWERS' COMMENTS

Reviewer #1 (Remarks to the Author):

The authors have addressed all my concerns.

Reviewer #2 (Remarks to the Author):

The authors have done a considerable amount of extra work, and I have no remaining concerns.